# The Molecular Basis for Zinc Bioavailability

**DOI:** 10.3390/ijms24076561

**Published:** 2023-03-31

**Authors:** Andrew G. Hall, Janet C. King

**Affiliations:** 1Department of Nutritional Sciences & Toxicology, University of California, Berkeley, CA 94720, USA; 2Department of Nutrition, University of California, Davis, CA 95616, USA

**Keywords:** zinc, bioavailability, absorption, phytate, protein, calcium, iron, albumin, metallothionein

## Abstract

Zinc is an essential micronutrient, and its deficiency is perhaps the most prevalent and least understood worldwide. Recent advances have expanded the understanding of zinc’s unique chemistry and molecular roles in a vast array of critical functions. However, beyond the concept of zinc absorption, few studies have explored the molecular basis of zinc bioavailability that determines the proportion of dietary zinc utilized in zinc-dependent processes in the body. The purpose of this review is to merge the concepts of zinc molecular biology and bioavailability with a focus on the molecular determinants of zinc luminal availability, absorption, transport, and utilization.

## 1. Introduction

As an essential micronutrient with a nearly ubiquitous presence in nature, zinc is needed for all known aspects of life. Based on the quantification of putative zinc protein binding domains, it is estimated that one-tenth of all human proteins require zinc as a structural element or for an enzyme active site [1]. The structural, catalytic, and regulatory roles of zinc provide the foundation for a broad array of basic cellular functions. Consequently, zinc deficiency affects numerous critical functions, including metabolic, immune, and neurological processes [2].

With zinc nutriture underlying a public health burden of communicable and non-communicable diseases [3,4,5,6], human zinc deficiency is estimated to be the most prevalent nutrient deficiency worldwide [7,8]. The etiology of human zinc deficiency has historically been attributed to diets with low zinc bioavailability, e.g., the proportion of dietary zinc available for zinc-dependent functions [9], with primary attention to diets low in zinc and high in phytic acid [10].

Although dietary factors that determine zinc absorption have been studied extensively [11,12], few studies have examined the factors that affect its ultimate utilization in zinc-dependent functions. Furthermore, since zinc’s numerous biological roles do not allow the use of isolated functions specifically attributable to zinc that can be readily measured in the clinical assessment of zinc status, the true prevalence of zinc deficiency remains largely unknown [13].

The molecular nature of zinc and its ligands determine the movement and function of zinc in the body, and they provide a mechanistic basis for understanding the pathology of zinc deficiency. However, most of the focus on reducing zinc deficiency has been to increase its bioavailability from diet or supplements with only a minimal exploration of the molecular determinants. The purpose of this review is to join recent advances in the understanding of the molecular nature of zinc metabolism with its bioavailability, specifically, how zinc’s unique chemistry determines its availability in the lumen for absorption, its transport and regulation in the body, and its utilization in numerous catalytic, structural, and regulatory roles.

## 2. Unique Chemistry of Zinc

Zinc is a redox-inactive divalent transition metal. Its biological activity is determined primarily through its ability to serve as a Lewis acid for catalysis or as a component of protein structures and superstructures and its coordination with amino acid side chains for regulating its various chemical and structural roles [14,15,16]. The bonds between zinc and the side chains of zinc-binding amino acids and between zinc and water thus determine the nature of its biological function.

The binding of zinc ions is unique compared to other transition metals. The coordination of aqueous zinc may include four, five, or six water molecules [14]. In its catalytic roles, zinc is coordinated between three amino acids (including one or more histidine residues) and one or more water molecules [17]. As with aqueous zinc, the zinc in active catalytic sites on proteins may take on a coordination number of four, five, or six. In contrast, structural zinc is frequently coordinated between four amino acids, including two or more cysteine residues. Steric repulsion by cysteine sulfur prevents zinc from forming bonds with water. This exclusion of water molecules maintains the neutrality of zinc.

Following the Irving-Williams series, divalent metal binding sites on metalloproteins have the highest preference, or affinity, for copper and zinc and the lowest preference for magnesium and calcium [18]. Free zinc readily displaces other divalent metals from their binding sites [19]. To prevent cytotoxic effects due to the mis-metalation of proteins, free metals are maintained at a concentration inversely proportional to their respective binding preference. Thus, cytosolic free zinc concentration is maintained in the picomolar range [20], more than three orders of magnitude lower than free magnesium or calcium, the two divalent metals with the lowest protein binding preference [19,21].

The natural reactivity of zinc in the presence of water, its ability to readily displace other divalent metals from their protein binding sites, and its broad functional importance as a catalytic and protein structural element require tightly regulated movement of zinc from its intestinal absorption through to its delivery to systemic cells for its numerous catalytic, structural, and regulatory roles [22]. To illustrate the extremity of this regulation of free zinc, consider the number of free zinc atoms indicated by picomolar concentration in the cytosol of a typical cell. Assuming that the cytoplasm makes up about half of the cell volume, and the cytosol makes up 61% of the cytoplasmic volume [23], about 30% of the cell volume is cytosol. Given a cell volume of 200 fL, and cytosolic free zinc ranging from 60 to 270 pM [24], the cytosol of a single cell would only contain between two and ten free zinc atoms.

Similar to copper, affinity gradients facilitate the controlled transfer of zinc [25]. As zinc moves through biological systems, it is transferred from one ligand to another, a process that is more rapid between ligands with similar binding affinities [26,27]. This controlled transfer of zinc is primarily mediated by a complex network of zinc carriers and transporters. However, unlike copper, zinc does not have a strong tendency towards a particular bond geometry [14], allowing more flexibility for zinc-binding sites to serve dual roles as zinc donors simply due to their relative zinc binding preference and proximity to an acceptor zinc-binding site. With this background regarding the unique chemistry of zinc and the nature of its movement in biological systems, the following section discusses the known molecular mechanisms underlying zinc bioavailability.

## 3. Zinc Bioavailability

Nutrient bioavailability is defined as the proportion of a nutrient in the diet utilized in the body [9]. Since this utilization is often difficult to quantify in a way that is traceable to diet, the concept of bioavailability is divided into three phases: availability for absorption in the intestinal lumen; absorption, retention, and distribution in the body; and tissue and cellular utilization [9]. While nutrient availability or absorbability may serve as estimates of bioavailability, zinc bioavailability is most correctly defined as the proportion of zinc in the diet that is ultimately utilized in zinc-dependent functions (Figure 1).

Bioavailability describes a dietary quality with a typical emphasis on the components that are the most readily measured, luminal availability, and absorption. However, this emphasis encourages the assumption that once zinc is absorbed, its chance of utilization is not further modified by dietary or physiological factors. In an article on nutrient bioavailability in humans and experimental animals, JS Godber cautioned that bioavailability should account for “intrinsic and extrinsic factors that may influence the ultimate utilization” [28]. In this review, we will explore the potential contributions of several extrinsic (e.g., the composition of diet) and intrinsic factors (e.g., the function of proteins regulating zinc metabolism) that may influence zinc bioavailability.

A related term, drug bioavailability, is defined as the proportion of dose that reaches the site of action in its active form [29]. A major distinction between nutrient and drug bioavailability is the presence of food. Zinc is naturally a component of food, although dietary zinc may be provided artificially as a fortificant or as orally administered supplement pills. The latter may be taken fasted or with food. Other therapeutic forms, such as those used for topical administration or injection, do not interact directly with the gastrointestinal tract or digestive processes that determine nutrient bioavailability. Thus, either or both definitions could apply to zinc, primarily depending on whether the purpose is nutritional or therapeutic. Here we emphasize nutrient bioavailability and discuss the roles of food components and normal digestive processes in enhancing or inhibiting zinc absorption, distribution, and utilization, although we also discuss data relevant to the therapeutic administration of zinc.

### 3.1. Zinc Luminal Availability and Absorption

Upon ingestion of a meal, digestive processes begin to free the dietary zinc from the food matrix, making it available for absorption. The acidic gastric environment hydrolyzes dietary zinc and, after passing the pyloric sphincter into the duodenum, pancreatic juices containing bicarbonate and digestive enzymes mix with the digestate, neutralizing the gastric acids and begin hydrolyzing dietary protein, and thereby releasing protein-bound dietary zinc. Pancreatic juices contain a substantial amount of zinc. Several milligrams of zinc are secreted daily in pancreatic juices [30,31], whereas only about 2.5 to 3.5 milligrams of zinc per day are absorbed from food to meet physiological requirements [32]. Plasma zinc kinetics following oral zinc administration are consistent with enterohepatic recirculation, whereby absorbed zinc is secreted with pancreatic juices and reabsorbed [33].

Dietary zinc is thus mixed with digestive juices and endogenously secreted zinc, components of food that have been made soluble through digestive processes and continue to be broken down, and those that remain insoluble. Reducing this to a molecular perspective of zinc before its absorption, zinc in the digestive matrix is bound to water molecules and numerous zinc-binding ligands of varying affinities.

In the context of digesting foods, soluble zinc-binding ligands, such as peptides and amino acids, help to maintain the solubility of zinc and thus promote its access to zinc transporters. When these ligands are less soluble (e.g., undigested protein or phytate complexed with calcium), they may be capable of precipitating zinc, thus preventing its contact with the active sites of zinc transport proteins. It is primarily the interaction of dietary zinc with water, zinc-binding ligands, and the zinc-binding sites on zinc transport proteins that determine the availability of zinc for absorption.

In the context of zinc-fortified foods or supplements, additional complications may be present. For example, when a zinc supplement or tracer is administered in the post-absorptive (fasted) state, the fractional absorption and appearance in plasma are much greater than normally observed with food [34,35]. This suggests that foods and/or digestive processes reduce zinc bioavailability. However, it is also known that zinc is rapidly drawn from plasma, predominantly directed to the liver, with the intake of food [36]. The consequence of postprandial zinc dynamics for the quantification of zinc absorption has not been examined in any study we are aware of.

Furthermore, direct comparisons of any marker of cellular zinc utilization, beyond plasma zinc concentration, when zinc is taken fasted vs. with food are startlingly few. We are aware of only one study, from our lab, where the same form and amount of zinc was given in the fasted state vs. with a breakfast meal. The effects on plasma zinc concentration and on a zinc-dependent cellular process lack congruence [37]. Although zinc taken in the fasted state daily for two weeks has a greater effect on plasma zinc concentration, the effect on zinc-dependent essential fatty acid desaturation is more pronounced when the zinc is taken with food.

In addition, the forms of zinc used for these purposes may have unique metabolic qualities atypical of zinc as a natural component of food. Consider zinc oxide, commonly used as a fortificant and in zinc supplements. Zinc oxide has low solubility and only releases zinc through a reaction that releases reactive oxygen species (ROS) [38]. Zinc given as supplements in the fasted state had a lower fractional absorption when provided as zinc oxide, compared with zinc citrate or zinc gluconate [39]. Some of the participants had undetectable (<1.5%) fractional absorption from zinc oxide, perhaps due to inadequate stomach acidity [40]. Cytotoxicity of ROS released from zinc oxide, even at levels typical of fortification, may also be of concern for gut health [41].

For the purpose of this review, we focus on the molecular basis of zinc bioavailability in the context of zinc as a natural component of food and normal zinc metabolism. We do, however, include limited discussion of data beyond this focus, where they would be beneficial toward understanding the molecular basis for zinc bioavailability.

Dietary factors that enhance or inhibit zinc absorption make up a complex and active area of study. In the following subsections, we discuss four factors that have been influential on observed zinc absorption based on the combined data of human tracer studies: the dietary content of zinc, phytate, calcium, and protein [12,42]. We will also comment on the effects of dietary iron, which was not found to be a significant factor in zinc absorption from food [12].

#### 3.1.1. Effects of Dietary Zinc

The absorption of dietary zinc is concentration dependent and saturable. Thus, it may be modeled using the Hill equation [34]. Zinc is absorbed over the entire length of the small intestine, from the duodenum to the distal ileum [43]. Its intestinal uptake is the most rapid in the proximal small intestine. Although peristaltic mobility and fluid transit time are known influences in intestinal absorption, especially with enteric recirculation [44], we are not aware that their contribution to zinc absorption or reabsorption has been measured.

Zip4, the zinc transporter primarily responsible for the absorption of dietary zinc from the digestive matrix, is localized to the apical membrane of luminal enterocytes. Several mutations in the gene encoding Zip4 cause acrodermatitis enteropathica (AE) [45,46], a rare condition characterized by severe zinc deficiency that is observed post-weaning in breastfed children or earlier in children who were not breastfed [47,48]. Zip4 transports free zinc via an elevator-type transport mechanism typical of members of the Zrt/Irt-like protein (ZIP) family, whereby two transport proteins, each with eight transmembrane domains, form a dimer and share a passage for zinc [49].

Recent studies regarding the mechanism of Zip4 action demonstrate the importance of its extracellular histidine-rich (M2) region and its affinity for zinc (Table 1). The M2 region of Zip4 enhances the efficiency of zinc transport by capturing zinc from the bulky aqueous solvent outside the cell and transferring it to the zinc transport site (M1) [50,51]. Once zinc binds to the M2 site, it is only released via transfer to M1 and subsequent transport into the cell [51].

Replacement of the histidine residues of the M2 site with serine prevents zinc binding to the M2 domain. Interestingly, this mutation does not reduce the overall affinity of Zip4 for extracellular zinc, although it reduced the maximal zinc transport rate (V_max_) by approximately 20% [50]. The M2 and M1 zinc binding sites of Zip4 thus work in concert, consistent with an affinity gradient that enhances the transfer of aqueous zinc from the digestate in the small intestine and across the apical membrane of luminal enterocytes.

A second Zip4 histidine-rich (M3M4) domain faces the cytoplasm. The M3M4 domain confers a zinc-sensing ability and, upon binding cytosolic zinc, triggers the endocytosis, ubiquitination, and proteolytic degradation of Zip4 [52,53]. It is this ability of Zip4 to sense increases in cytosolic zinc and aggressively downregulate zinc absorption through the removal of Zip4 that contributes to the distinctly saturable nature of zinc absorption and prevents cytotoxicity due to the overabsorption of dietary zinc.

Zip4 is recovered following transcriptional regulation mediated by Krüppel-like factor 4 [54]. In murine models of zinc deficiency, increases in Zip4 transporter expression and localization to the plasma membrane and zinc absorption have been observed in response to low zinc intake [55,56]. However, structural motifs responsible for zinc sensing differ between murine and human Zip4 [22].

Although currently available data do not rule out the upregulation of zinc absorption in humans, tracer data indicate that it is primarily in response to factors other than low zinc intake. Following a period of severe dietary zinc depletion in men, zinc balance was conserved via the reduction of endogenous zinc excretion [57], and a compensatory increase in zinc absorption was not observed following a previous period of low zinc intake [58]. However, fractional zinc absorption increases in response to normal pregnancy [59].

The molecular properties of human Zip4 are thus consistent with kinetic models derived from zinc tracer studies [60]. At low concentrations of soluble zinc in the digestate, Zip4 efficiently and actively transports dietary zinc into luminal enterocytes via two zinc-binding sites working in concert. As the total amount of zinc absorbed into cells increases, the internalization and removal of Zip4 slows and eventually halts further zinc absorption.

#### 3.1.2. Effects of Dietary Phytate and Calcium

Components of the diet that would chelate zinc or otherwise prevent contact with Zip4, for example, by reducing zinc solubility, may reduce zinc absorption. This is most frequently observed as a result of dietary phytate, or myo-inositol hexakisphosphate, the storage form of phosphorus in plants. As dietary phytate increases, zinc absorption decreases [60]. Of the divalent metal nutrients, phytate binds zinc and copper with the highest affinity, though it also binds iron, calcium, and magnesium. Complexes of phytate and apparent dissociation constants of selected zinc ligands.

**Table 1 ijms-24-06561-t001:** Apparent dissociation constants of selected zinc ligands.

Type	Zinc Ligand	Dissociation Constant (K_d_) ^a^	pK_d_ (−log K_d_)	Ref.
**Protein zinc-binding** **domains**	Zip4 M2 region	2.9 × 10^−5^ M	4.5	[50]
	Zip4 M1, M2 average	6.2 × 10^−7^ M	6.2	[50]
	Zip4 M3M4 region	6.0 × 10^−9^ M	8.2	[53]
	MT, α domain	2.5 × 10^−13^ − 3.2 × 10^−14^ M	12.6 − 13.5 ^b^	[61]
	MT, β domain	4.0 × 10^−12^ − 6.3 × 10^−13^ M	11.4 − 12.2 ^c^	[61]
	Albumin	4.0 × 10^−7^ M	6.4 ^d^	[62]
	α-2-macroglobulin	8 × 10^−7^ M	6.1 ^e^	[63]
	CA2	8.0 × 10^−13^ M	12.1	[64]
	SOD1	4.2 × 10^−14^ M	13.4	[65]
**Amino acids and** **peptides**	Cysteine	9.8 × 10^−12^ M^2^	11.0	[14]
	Histidine	3.1 × 10^−9^ M^2^	8.5	[14]
	Aspartic acid	4.1 × 10^−6^ M^2^	5.4	[14]
	Glycine	1.3 × 10^−5^ M^2^	4.9	[14]
	Glutamic acid	2.2 × 10^−5^ M^2^	4.7	[14]
	Cys-Gly	1.5 × 10^−6^ M^2^	5.8	[66]
	Gly-His	1.4 × 10^−4^ M^2^	3.9	[66]
	Gly-Gly-His	1.6 × 10^−4^ M^2^	3.8	[66]
	Gly-Cys-Glu ^f^	2.3 × 10^−4^ M	3.6	[66]
**Other**	Clioquinol ^g^	7.9 × 10^−17^ M^2^	16.1	[67]
	EDTA	2.3 × 10^−14^ M	13.6	[14]
	Citric acid	1.2 × 10^−12^ M^2^	11.9	[14]
	Picolinic acid	1.3 × 10^−12^ M^2^	11.9	[14]
	D-penicillamine	2.8 × 10^−12^ M^2^	11.6	[14]
	Pyrithione	5.0 × 10^−12^ M^2^	11.3	[14]
	Phytic acid	3.7 × 10^−11^ M	10.4	[14]
	PBT2 ^g^	7.1 × 10^−7^ M^2^	6.2	[68]
	Folic acid	2.2 × 10^−6^ M^2^	5.7	[14]

Abbreviations: CA2, carbonic anhydrase 2; SOD1, copper/zinc superoxide dismutase; EDTA, ethylenediaminetetraacetic acid; L, ligand; M, molar; MT, metallothionein; PBT2, 5,7-dichloro-2-[(dimethylamino) methyl]-8-hydroxyquinoline. ^a^ Stoichiometry of ZnL is indicated by units of M, and ZnL_2_ by units of M^2^. ^b^ MT α domain pK_d_ values are 12.6, 12.7, 13.2, and 13.5. ^c^ MT β domain pK_d_ values are 11.4, 11.7, and 12.2. ^d^ Others have reported pK_d_ values of 7.1 [69] and 7.5 [70]. ^e^ Refers to the highest affinity zinc binding site. ^f^ Glutathione. ^g^ Clioquinol (5-chloro-7-iodo-8-hydroxyquinoline) and PBT2 are examples of dihalo-hydroxyquinoline drugs.

Zinc remains soluble in the digestive tract, while calcium and magnesium complexes with phytate are less soluble and readily precipitate from solution at the intestinal pH [71]. The interactions of dietary phytate with other components of the chyme, including divalent metals, protein, as well as amino acids and peptides released in protein hydrolysis, are complex and influence phytate solubility and potency in chelating dietary zinc [72].

Animal studies demonstrated that adding calcium as calcium carbonate to a high-phytate diet reduced zinc absorption [73,74]. However, this was not supported in a human tracer study designed to measure the effect, where calcium was provided by foods naturally high in calcium or commercially fortified with calcium [75]. Although the calcium fortificant was not stated, calcium carbonate is the most common calcium salt used in fortification in the U.S., where this study was conducted [76]. Moreover, increasing the calcium content of phytate-containing soy milk formula from 550 mg/L to 1340 mg/L increased zinc absorption from 14% to 20% [77]. Combining data from several human zinc absorption studies, Miller and colleagues observed a small but significant effect of calcium to increase zinc absorption, an effect that was pronounced in the presence of dietary phytate (Figure 2) [12].

Some have proposed, based on titration studies, that calcium could increase or decrease zinc solubility in the presence of phytate, depending on solute load and the molar ratios of phytate, zinc, and calcium [72,78]. For example, calcium (as calcium chloride), when added in molar excess of phytate to a solution of phytate-bound zinc (six phytates to one zinc), was capable of releasing 35% to 40% of the zinc from the phytate [79]. A similar effect of increased zinc absorption in adults was observed when calcium was added to phytate-containing soy formula as calcium chloride [77].

None of the studies included in Miller’s 2013 model used supplements or obvious fortification as sources of the modeled nutrients. It is possible that another dietary component or quality that changed with calcium contributed to oral zinc tracer absorption. However, the molar ratios of the constituents in the diets included in Miller’s analysis were consistent with the titration experiments and human data that replicated this effect: zinc (0.066–0.32 mmol/d), phytate (0.37–5.65 mmol/d), and calcium (10.7–31.2 mmol/d) [12]. Taken together, these data suggest that the speciation of calcium in the chyme, and its molar ratios with phytate and zinc, are important factors in determining the effects of dietary calcium on zinc bioavailability.

#### 3.1.3. Effects of Dietary Protein

Protein intake increases zinc absorption (Figure 2) [12]. However, the speciation of zinc with dietary protein and peptides released in the digestive process, as well as with digestive enzymes and their remnants, is complex, and molecular mechanisms that would explain these observations are unknown. Early studies demonstrated that in vitro hydrolysis of oil seed meals using purified digestive enzymes increased zinc bioavailability in chicks [80,81]. Phosphopeptides produced from the in vitro digestion of casein also increased zinc absorption in suckling rats and increased zinc uptake by cultured human enterocytes (Caco-2 cells) [82].

Pancreatic secretions are the primary source of digestive proteases which break down proteins to their constituent amino acids and peptides for absorption [83]. Pancreatic enzymes themselves have been observed to mediate zinc absorption. In children with cystic fibrosis who were receiving pancreatic enzyme replacement therapy, withholding the replacement enzymes for one day reduced fractional zinc absorption by nearly one-third, from 50% to 38% [84].

Several amino acids, i.e., glutamate, glycine, histidine, tryptophan, and dipeptides, including Leu-His and Gly-Gly, increased zinc absorption in the perfused ileum of rats [85]. Furthermore, the addition of a molar excess of these ligands led to competitive inhibition and a marked reduction in zinc absorption. In a transport model of zinc absorption in an isolated brush border membrane, the Gly-Gly-His tripeptide also increased zinc absorption, and competitive inhibition of zinc absorption, at a molar excess of the peptide over zinc, was similarly observed [86]. In humans, providing 20 mg of zinc as zinc histidinate had a greater effect on serum zinc concentration than the same amount of zinc as zinc sulfate when both were given in the fasted state [87]. Measured (apparent) affinities of selected amino acid or peptide complexes with zinc are presented in Table 1.

It has been proposed that zinc may be co-absorbed via amino acid transporters based on the observation of competitive inhibition by excess amino acids [85,86] and the similar absorption of zinc with amino acids in cultured enterocytes from an AE patient compared with control [88]. However, none of these studies have demonstrated that a zinc amino acid complex remains intact while transiting across the plasma membrane.

Several bioactive peptides, which are resistant to hydrolysis and too large to be transported via amino acid transporters, are also capable of increasing the cellular uptake of zinc from cell culture media [89,90,91]. Alternatively, it is plausible that soluble zinc-binding ligands in the chyme, within an appropriate range of affinities for zinc, would increase zinc absorption by creating an affinity gradient towards zinc delivery to the active sites of zinc transporters [27].

Data from the study of AE patients further reveal how dietary zinc may be absorbed without functional Zip4. AE may be treated successfully with supplemental zinc, perhaps due in part to passive transport at high doses. However, alternative pathways of zinc absorption are active at much lower levels of zinc. For example, in cell models of zinc absorption, a zinc-chelating media is needed to prevent background absorption by non-Zip4-mediated pathways [51]. Although it has been suggested this could represent non-carrier-mediated or passive transport, a passive component to zinc absorption is not supported by kinetic data from human zinc tracer studies at doses up to 30 mg of zinc in adults (approximately three to four times the Estimated Average Requirement) [34,92].

Zinc absorption may also be induced in AE patients with the administration of milk [93], suggesting that component(s) or properties of milk increase zinc absorption by pathways other than Zip4. Citrate [94] and picolinate [95] were both proposed, though not conclusively demonstrated, as the components of milk responsible for this effect. When zinc was provided as zinc dipicolinate, the dose of elemental zinc needed to prevent the return of symptoms in each of the three AE patients was reduced to about one-third, compared with zinc sulfate [96]. The patients, aged 4 to 16 years, required 30 to 45 mg zinc/d as zinc sulfate and 10 to 15 mg zinc/d as zinc dipicolinate to reliably avoid the return of a skin rash. However, milk also contains proteins, peptides, and essential fatty acids that may conceivably contribute to zinc absorption. Understanding the mechanisms and their relative contributions to zinc absorption with AE would require further study.

Dihalo-hydroxyquinoline drugs, potent zinc ionophores that form hydrophobic complexes with zinc, also induce zinc absorption in AE patients [97]. The amount of oral zinc needed for the treatment of a mouse model of AE is also reduced when zinc is provided with the dihalo-hydroxyquinoline drug clioquinol [98]. Several hydrophobic zinc ionophores are capable of delivering zinc across the cell membrane without a transporter, a property that has attracted attention to their potential therapeutic use in the treatment of infections, cancer, and neurological diseases [99,100,101,102]. Similarly, zinc complexes with EDTA cross the apical membrane of luminal enterocytes intact. In contrast, zinc-EDTA did not cross the basolateral membrane in an animal model [103]. The enhancement of zinc absorption with EDTA may, rather, be due to the interaction of EDTA with complexes of phytic acid, calcium, and magnesium that would otherwise inhibit zinc absorption [11,104].

Peptides that resist hydrolysis by digestive proteases also tend to be hydrophobic [105,106], whereby they would associate with lipids in the digestate. Lipophilic zinc amino acid complexes have also been demonstrated to promote antibiotic activity via the direction of zinc across the bacterial cell wall [107]. Although data supporting this mechanism of zinc absorption are sparse, the transport of hydrophobic zinc complexes across the plasma membrane may have a limited contribution to dietary zinc absorption. Several bioactive peptides that increase zinc absorption are hydrophobic [89], a property that may allow zinc-peptide cotransport with lipids. Furthermore, the intragastric administration of essential fatty acids increased the absorption of an oral zinc tracer in a dose-dependent manner in neonatal rats [108].

Another putative pathway of dietary zinc absorption is mediated by ZnT5 splice variant B (ZnT5B). Initially designated hZTL1 [109], ZnT5 is encoded by the gene SLC30A5 [110]. While ZnT5, splice variant A, forms a heterodimer with ZnT6 on the surface of the Golgi complex, implicated in the metalation of alkaline phosphatase [111], splice variant B is localized to the apical membrane of small intestine enterocytes [112,113]. Using an antibody that could recognize either splice variant, we observed jejunal ZnT5 localized to the apical membrane of the villus crypt area in juvenile rats [114].

A zinc supplement (25 mg zinc as zinc sulfate) taken with food daily for two weeks led to a decrease in ZnT5 and Zip4 proteins, both apically localized, in ileal biopsies collected from human volunteers [115]. A Caco-2 cell model of zinc absorption that overexpressed ZnT5B on the apical membrane accumulated zinc in the cytoplasm, demonstrating ZnT5B capable of zinc transport into the cell [116]. These results are somewhat surprising, given that members of the ZnT family of zinc transporters are generally observed moving zinc in the direction out of the cytosol, whether into zinc-containing intracellular vesicles, such as with ZnT2 and ZnT4 or across the basolateral membrane into circulation, as with ZnT1 [22]. The molecular mechanism whereby ZnT5B could accomplish luminal zinc import warrants further investigation.

Key differences in the structure and regulation of Zip4 and ZnT5B reveal a divergence in their potential contributions to zinc homeostasis. With high intracellular zinc, Zip4 is regulated by internalization and proteolytic degradation when its cytoplasmic zinc sensing domain binds zinc, while ZnT5 is transcriptionally regulated by metallothionein (MT). The overexpression of MT 2A, or elevated extracellular zinc concentrations in normal cells, both repressed the SLC30A5 promotor, resulting in the downregulation of ZnT5 mRNA [110].

Zip4 has two zinc-binding sites that work in concert to increase its efficiency in capturing and delivering dietary zinc to its active transport site. It is not known whether ZnT5B may be active in transporting aqueous zinc from the chyme or whether zinc-binding ligands, such as those released in the digestion of protein, would enhance the absorption of dietary zinc via ZnT5B. Applying the concept of zinc affinity gradients, the potential of ZnT5B to import dietary zinc in concert with zinc-binding ligands in the chyme, including those resulting from protein digestion, presents a reasonable area for future exploration.

#### 3.1.4. Effects of Dietary Iron

Miller and colleagues found that dietary iron primarily from natural food sources had no significant effect on zinc absorption [12]. However, earlier studies reviewed by others [117] demonstrated that iron (as iron sulfate dissolved in water) decreased zinc absorption in adult volunteers when it was given following an overnight fast at Fe:Zn molar ratios of 1:1 up to 25:1 [118,119]. This inhibition is likely due to competition for binding to Zip4, i.e., an overwhelming concentration of divalent (ferrous) iron partially blocks the active (zinc-binding) site of Zip4. Data from cell studies show that human Zip4 has a greater specificity for zinc binding compared with iron. However, Zip4-mediated transport of a zinc tracer was reduced by 40% in the presence of excess ferrous iron [120].

In contrast, there was no effect of iron on zinc absorption in adults when the iron was supplied with food or as a fortificant, even at a 25:1 or 50:1 molar ratio for iron to zinc [118,119,121,122]. More recent studies in infants [123,124,125] and children [126,127] also found no effect of iron, at amounts consistent with fortification, on zinc absorption when the iron was supplied with infant formula or food, respectively. Sandstrom et al. further demonstrated that adding histidine (at a 100:1 molar ratio with the 40 µmol, or 2.6 mg, zinc) attenuated the inhibitory effect of iron on zinc absorption from water [119]. It is conceivable that the speciation of zinc in the complex milieu of zinc-binding ligands exposed through digestive processes protects zinc absorption from interference by moderate amounts of iron.

In some cases, the dietary iron content may correlate with zinc absorption when the differences in zinc absorption are due to other factors. For example, in the raw data of Miller et al., there was a significant correlation between iron intake and zinc absorption before including calcium and protein in the model [12]. In addition, as with zinc, plant-derived foods that are high in iron tend to be high in phytate [128]. It is thus possible that, without accounting for the phytate content of a diet, increased iron from plant sources could appear (erroneously) to have an inhibitory effect on zinc absorption, that is, in fact, due to increased phytate.

Consistent with the data from studies that provided iron as a beverage following an overnight fast, prenatal supplements containing 60 mg iron as iron sulfate and 250 µg folic acid reduced fractional zinc absorption by more than half when given to pregnant women in the fasted state. Fractional zinc absorption was ~20% with the prenatal supplement and 47% without any supplement [129]. This study design did not rule out the possibility that the presence of folic acid (pteroylglutamic acid) with iron contributed to the inhibition of zinc. Data describing the conditions whereby folic acid may have a substantial and repeatable inhibitory effect on zinc absorption are, however, lacking.

The only zinc tracer study conducted in human subjects found no effect on zinc absorption of 144 µg folic acid, provided as a fortified bread, on zinc absorption, even though the study was powered to detect a difference in zinc absorption greater than 8% [130]. Although folic acid can bind zinc, its affinity for zinc is comparable to that of glycine for zinc (Table 1). Therefore, as with iron, as long as the folic acid is taken with food, it is not likely to affect zinc absorption at the levels normally used in fortification.

#### 3.1.5. Summary of Zinc Absorption

Taken together, data presented in this section support a molecular explanation of the main components of diet known to affect human zinc absorption: zinc concentration, phytate, protein, and calcium contents. Models based on human tracer data that include these variables within a range of their natural concentrations in the diet explain 88% of the variability in dietary zinc absorption [12]. Data from in vitro and in vivo studies provide additional detail on the underlying mechanisms.

Zip4 is responsible for most dietary zinc absorption from a mixed diet, and the molecular behavior and regulation of Zip4 are consistent with kinetic observations of zinc absorption. However, the speciation of zinc also influences its absorbability, and other pathways, including the apically localized ZnT5B zinc transporter, co-transport with amino acids, or movement across the plasma membrane with hydrophobic zinc ionophores, have the potential to contribute to zinc absorption (Table 2). More study is needed on how the speciation of zinc in chyme may thus determine a preference for alternative zinc absorptive pathways.

### 3.2. Zinc Utilization

Absorbed zinc may be directed to utilization in cellular compartments, transported into the urine, or excreted back into the digestive tract. Balance studies indicate a positive linear relationship between zinc absorption and excretion [32,92]. Since there are no body stores for labile zinc readily released in response to low intakes, absorbed zinc that is not utilized in zinc-dependent functions is, in turn, excreted. The more zinc absorbed, the more that is excreted back out of the body. In addition to the total amount of zinc absorbed, little is known of factors that would determine the retention and utilization, versus the excretion, of absorbed zinc.

A review of the proteins that regulate zinc metabolism at the cellular, tissue, and whole-body levels, and their properties, however, may provide important insight into the nature of a zinc utilization axis. Zinc carriers and numerous zinc transporters maintain zinc homeostasis via the regulation of cell and organelle-specific transfer of zinc across membranes and are reviewed by others [22,26,131,132]. Here, we focus on metallothionein (MT) and albumin (Table 1), two carrier proteins with significant importance to the distribution and cellular utilization of dietary zinc.

#### 3.2.1. Metallothionein

MT is a compact, cysteine-rich protein highly conserved in nature [133]. Mammalian MTs contain 61 to 68 amino acids depending on the isoform, 20 of which are cysteine and are capable of binding up to seven zinc atoms. The MT α domain has four zinc sites with picomolar affinities (Table 1), while the β domain has three zinc sites with nanomolar affinities [61,134]. The level of intermediate zinc metalation of MT determines its function. As MT takes on zinc, its affinity for additional zinc decreases, suggesting a shift from sequestering zinc (where the zinc would be bound too tightly to exchange with other proteins readily) to zinc donation to enzyme active sites [61].

Due to its range of zinc-binding affinities, MT may moderate the metalation of numerous zinc-dependent apo-enzymes. This is exemplified by carbonic anhydrase 2 (CA2), a zinc-dependent enzyme responsible for the hydration of carbon dioxide to carbonic acid for transport in the blood or towards the production of hydrochloric acid in the stomach. MT donates zinc to the catalytic site of CA2, which has a pK_d_ (affinity) for zinc of 12.1 [64]. Only the last two zinc-binding domains of MT bind zinc with a lower affinity, with pK_d_ of 11.4 and 11.7. Therefore, only when MT carries a total of six or seven zinc atoms may it donate zinc to the active site of apo-CA2 [135]. However, with a zinc influx into the cell, apo-CA2 effectively competes with MT for zinc once MT carries more than three zinc atoms [61], suggesting that transient influxes of zinc are advantageous towards the metalation of CA2.

The role of MT in regulating the cellular response to oxidative stress may be additionally important in determining the direction of zinc toward utilization. MT neutralizes ROS through the oxidation of its cysteine residues [136]. This causes the release of zinc and polymerization of MT [137], which signals a further cellular response to include increased expression of MT [138]. In the presence of ROS, zinc is more readily released from the lower affinity β domain of MT [139].

Thus, an increased demand on MT to respond to ROS would be expected to lead to a concurrent shift in zinc metabolism, reducing zinc transfer to protein active sites, increasing the movement of zinc out of the cytosol, and, with the further expression of apo-MT, increased sequestration of cytosolic zinc to its high-affinity zinc-binding sites. Although it has implications for altered zinc homeostasis and reduced zinc utilization in conditions of oxidative stress, to our knowledge, this has not been explored in metabolic studies of zinc.

#### 3.2.2. Albumin

Albumin is the primary zinc carrier protein in blood plasma, where it mediates zinc transport between tissues. ZnT1, localized to the basolateral membrane of luminal enterocytes, is responsible for the transport of absorbed dietary zinc into the portal blood [131], where albumin collects the zinc for transport to the liver. Without albumin, zinc is not transported out of mucosal cells into circulation [140].

About 80% of the zinc in blood plasma is bound to albumin, which has a single high-affinity zinc-binding site [141]. Most of the remaining zinc in plasma is bound to the protease inhibitor, α-2-macroglobulin [142], which has an affinity for zinc similar to that of albumin [63]. Under normal conditions in the healthy individual, albumin is at a molar ratio with zinc of approximately 40:1, conferring a high capacity of blood plasma to scavenge free zinc [2]. The affinity of albumin for zinc, pK_d_ of 6.4 [62], combined with its molar excess over zinc, effectively keeps free zinc in plasma in the range of 1–3 nM [143].

Albumin is also the carrier protein for free fatty acids (FFA). In the lipolytic release of FFA from circulating chylomicrons, albumin is needed for clearance of FFA from lipoprotein lipase and transport to the receptors mediating cellular FFA uptake [144,145]. In support of this function, the zinc-binding site on albumin serves as the primary FFA binding site [146]. As FFA are released into circulation with lipolysis, zinc is released from albumin and thus directed into cellular compartments.

Food intake directs plasma zinc into slow turnover tissue stores that are predominantly localized to the liver [36], consistent with its utilization, an effect mediated by postprandial lipolysis. Due to this postprandial cross-talk between zinc and FFA, food intake determines zinc utilization. This is consistent with our observations in apparently healthy men randomized to take a 25 mg zinc supplement (as zinc gluconate) in the fasted state 30 min before breakfast or with their breakfast meal. The effect on essential fatty acid desaturation was greater when the zinc was taken with food [37].

#### 3.2.3. Summary of Zinc Utilization

In exploring the interaction of zinc with its carrier proteins, potential mechanisms underlying an axis of zinc utilization vs. clearance are revealed. Varying affinities of MT zinc sites determine the direction of zinc to the metalation of zinc enzymes, which differs between steady state and conditions of cellular zinc influx. The dual roles of MT in carrying zinc and responding to oxidative stress are interactive in regulating the distribution of cellular zinc. Likewise, the dual roles of albumin in carrying zinc and fatty acids are interactive in regulating postprandial zinc utilization.

## 4. Conclusions

More than 50 years have passed since John Reinhold proposed that reduced zinc absorption due to dietary phytate causes human zinc deficiency [147]. Since then, considerable progress has been made toward understanding the bioavailability of dietary zinc and the molecular mechanisms facilitating zinc metabolism. However, the prevention of zinc deficiency remains challenging.

This review of molecular mechanisms supporting dietary zinc bioavailability points to a need for further research in four areas: the role of the complex speciation of zinc in the digestive matrix in its absorption, the mechanisms for recycling secreted zinc to sustain metabolic functions, the impact of dietary and physiological factors in altering the speciation and transfer of zinc within blood plasma and cellular compartments, and the distribution of absorbed zinc to tissue utilization versus excretion.

An expansion of scientific knowledge in these areas is needed to advance clinical care and public health solutions for zinc deficiency worldwide. This scientific advancement depends on merging molecular and whole-organism approaches in studies of zinc metabolism.

## Figures and Tables

**Figure 1 ijms-24-06561-f001:**
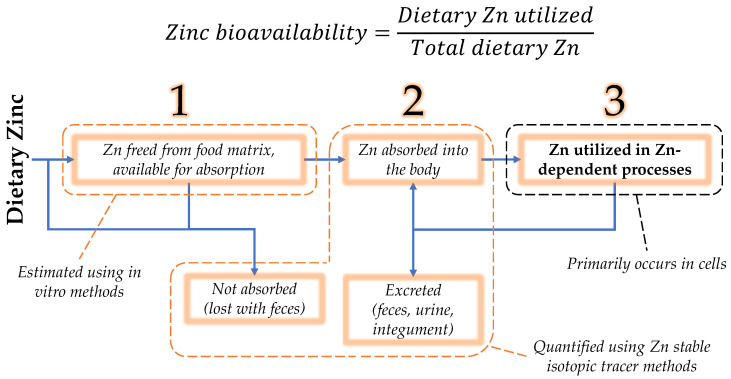
Zinc bioavailability is the dietary zinc that is utilized in zinc-dependent processes, expressed as a fraction or percent of the total dietary zinc. Towards its estimation, bioavailability may be separated into three phases: (1) luminal availability, (2) absorption/retention/distribution in the body, and (3) utilization [9]. Zinc that is absorbed may be retained for utilization, secreted endogenously, as a component of pancreatic juice, or directly by intestinal cells. Secreted zinc may be reabsorbed or excreted in the feces.

**Figure 2 ijms-24-06561-f002:**
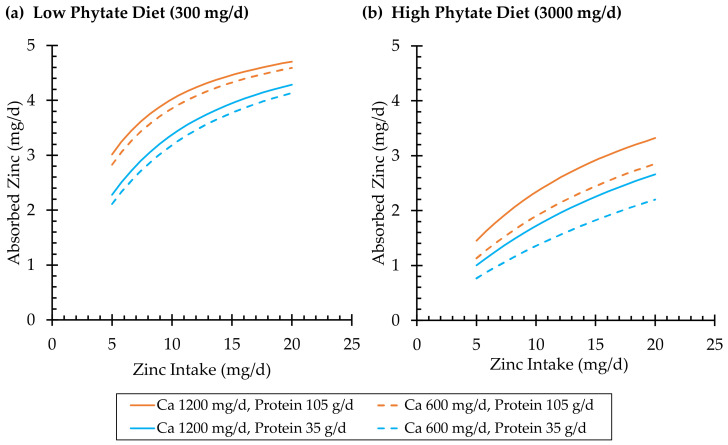
Effects of dietary zinc, phytate, calcium, and protein on zinc absorption, predicted by Miller’s 2013 model [12]. Each curve shows the predicted zinc absorption versus the dietary zinc intake by phytate content (charts (**a**) vs. (**b**)), calcium content (solid vs. dashed lines), and protein content (orange vs. blue lines). Values for zinc, phytate, calcium, and protein are within the ranges covered by data used to develop the model, as stated in Table 1 of the referenced study.

**Table 2 ijms-24-06561-t002:** Pathways of intestinal zinc absorption.

Pathway	Evidence of Contribution to Dietary Zinc Absorption
Zip4	Mutations resulting in loss of Zip4 function cause severe zinc deficiency [46]. The molecular mechanism of Zip4 transport and cellular regulation [50,53] is consistent with the kinetics of zinc absorption measured in humans [34].
ZnT5B	Apically localized on brush border membrane enterocytes [113,114,115], ZnT5B increases zinc absorption in cultured enterocytes [116]. In conditions of excess zinc or overexpressed MT, ZnT5 is transcriptionally downregulated [110,112]. Zinc supplementation in humans causes a reduction in Zip4 and ZnT5B protein measured in ileal biopsies [115].
Co-transport with amino acids	Initially proposed based on observations that zinc-binding amino acids increase zinc absorption when in molar parity with zinc and competitively inhibit zinc absorption when in molar excess of zinc [85,86]. Zinc complexed with amino acids is absorbed by cultured enterocytes lacking Zip4 [88].
Co-transport with hydrophobic peptides or zinc ionophores	Hydrophobic zinc ionophores induce zinc absorption [97] and reduce the demand for therapeutic zinc in patients lacking Zip4 [96,98]. Hydrophobic zinc-binding peptides increase zinc uptake from cell culture media [89,90,91]. Intragastric administration of essential fatty acids leads to a dose-dependent increase in zinc tracer absorption in neonatal rats [108].
Passive transport	High-dose zinc supplementation induces zinc absorption in patients lacking Zip4. However, interaction with alternative zinc absorptive pathways was not ruled out. Kinetic studies find no evidence of passive absorption at zinc intakes below 30 mg Zn/d [34].

## Data Availability

Not applicable.

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
