# Peer review of "The Molecular Basis for Zinc Bioavailability"

_ijms, 2023, doi:10.3390/ijms24076561_

Round 1

Reviewer 1 Report

Review of manuscript IJMS-2280270 for International Journal of Molecular Sciences

The manuscript by Hall and King reviews the basis of our knowledge of zinc bioavailability over the last 50 years. In all the topic is of major interest and the manuscript is well written and deserves publication. However, some minor points would improve the quality of the manuscript.

1.      From the title someone would assume, that information regarding the bioavailability from different zinc supplements will be described. Therefore, a table should be included with information available for the differences in bioavailability of different zinc compounds. At least the most often used compounds should be listed like ZnO, ZnSO4, ZnCl, Zn-citrate, Zn-gluconate, Zn-histidine, Zn-aspartate, Zn-picolinate, Zn-orotate, Zn-acetate and others.

2.      From table 1, the listed compounds and parts of the text, someone would get the impression, that zinc bioavailability is just a matter of dissociation constants. However, for example EDTA was used for decades in feed for farm animals to increase zinc uptake. This should be at least discussed. Furthermore, zinc citrate is the main compound in breast milk, which as described by the authors bypasses Zip4 absence in AE, since symptoms start post weaning.

3.      The paragraph on amino acids (lines 256ff) should be enlarged, since more is known about specific amino acids which increase zinc uptake. Some amin

4.      In the conclusion it may be helpful to mention that in contrast to NIH, the EFSA and some European state authorities (e.g. DGE) included phytate intake into the recommendations for zinc uptake to consider reduced zinc bioavailability due to phytate.

5.      Lastly it may be interesting to mention the recent release of a free Zinc-APP to calculate zinc uptake in relation to food intake and phytate concentration (Trame et al. 2023 J. Trace Elem. Med. Biol.), since this is not easy for consumers.

In conclusion, the manuscript is acceptable after minor revision.

Author Response

Reviewer 1:
The manuscript by Hall and King reviews the basis of our knowledge of zinc bioavailability over the last 50 years. In all the topic is of major interest and the manuscript is well written and deserves publication. However, some minor points would improve the quality of the manuscript.

Thank you for your comments. We have responded point-by-point below. 

1.      From the title someone would assume, that information regarding the bioavailability from different zinc supplements will be described. Therefore, a table should be included with information available for the differences in bioavailability of different zinc compounds. At least the most often used compounds should be listed like ZnO, ZnSO4, ZnCl, Zn-citrate, Zn-gluconate, Zn-histidine, Zn-aspartate, Zn-picolinate, Zn-orotate, Zn-acetate and others.

Several of the listed zinc complexes are already in Table 1. Adding a table comparing the bioavailability of these different compounds, while interesting, would not greatly contribute to the mechanistic understanding presented in this paper. Furthermore, a comparison of supplement bioavailability would require consideration for zinc utilization whether the supplements are taken in the fasted state or with food. Currently, the supporting data in this area are too limited to do so. We have added explanation to the text. 

2.      From table 1, the listed compounds and parts of the text, someone would get the impression, that zinc bioavailability is just a matter of dissociation constants. However, for example EDTA was used for decades in feed for farm animals to increase zinc uptake. This should be at least discussed. Furthermore, zinc citrate is the main compound in breast milk, which as described by the authors bypasses Zip4 absence in AE, since symptoms start post weaning.

Similar to other zinc ionophores, zinc complexes with EDTA cross the apical membrane of luminal enterocytes intact. However, zinc-EDTA did not cross the basolateral membrane in an animal model. The enhancement of zinc absorption with EDTA observed in livestock may, in contrast, may be due to interaction of EDTA complexes of phytic acid, calcium, and magnesium that would otherwise inhibit zinc absorption …  Also, citrate and picolinate were both proposed as the components of milk responsible for recovery of zinc absorption in AE. Milk also contains proteins, peptides, and essential fatty acids that could conceivably contribute to this effect. The mechanisms of zinc absorption and their relative contributions to the role of milk’s protective effect with AE have not been studied. We noted that providing zinc as zinc dipicolinate to AE patients reduces the demand for therapeutic zinc supplementation by >60%. 

3.      The paragraph on amino acids (lines 256ff) should be enlarged, since more is known about specific amino acids which increase zinc uptake. Some amin

The paragraph presents selected examples to illustrate the effect of amino acids and peptides on zinc absorption. It was not intended to be comprehensive, rather to support exploration of the possible molecular mechanisms behind this effect. 

4.      In the conclusion it may be helpful to mention that in contrast to NIH, the EFSA and some European state authorities (e.g. DGE) included phytate intake into the recommendations for zinc uptake to consider reduced zinc bioavailability due to phytate.

Although the concepts presented in this paper have implications for recommendations and standards in human nutrition, their mention would beg a more detailed explanation beyond the scope of this paper. We do, however, plan to address this in another manuscript. 

5.      Lastly it may be interesting to mention the recent release of a free Zinc-APP to calculate zinc uptake in relation to food intake and phytate concentration (Trame et al. 2023 J. Trace Elem. Med. Biol.), since this is not easy for consumers.

Thank you for letting us know about this app! Again, although interesting, this is beyond the scope of the current paper. 

In conclusion, the manuscript is acceptable after minor revision.

Reviewer 2 Report

Thank you for your work in zinc nutrition.  Appreciate the thorough review of an important physiologic topic.  The systemic bioavailability of zinc remains an important consideration both in public health and clinical therapy.  I offer some minor comments/suggestions for your consideration that may benefit the reader. 

General Comments

1. Consider whether a brief description of the complexity of compartmental analysis for overall zinc kinetics would provide the reader some additional context for appreciating your review of bioavailability.

2. Consider a brief discussion about zinc supplements, of the varying salts, with different potential bioavailability, which readers may appreciate.  The physiologic/pharmacologic concept of bioavailability remains similar: taking into account factors at sequential steps from pre-absorption (eg solubility), the absorption process across the enterocyte’s apical membrane (eg, transporters), disposition within enterocyte (eg, storage, efflux), post-absorption from basolateral membrane (eg, transporters, carriers, and distribution to sites of function or excretion).  May require reference to paracellular absorption at pharmacologic dosing (commonly 50 mg elemental zinc up to 2-3 times daily).

3. Address limited roles of DMT1, ZIP10, ZIP11??

 Introduction

· Line 24 – can remove “a” (…zinc deficiency affects numerous critical functions…)

· L 27-28 – readers unfamiliar with nutrition status surveys may find confusing given the subsequent statement in Line 36-37 re: Zn deficiency

· L 40-41 – good point about the focus to increase Zn [bio]availability; consider mentioning both diet and supplementation

 Unique chemistry of zinc

· L 69 – can remove the first mention of “binding” (…metals with the lowest protein binding preference.”)

· L 72-73 – consider: “…movement of DIETARY OR SUPPLEMENTED zinc from its GUT absorption through to its delivery to SYSTEMIC cells…”

· L 79-80 – appreciated this illustrative example

· L 84 – using “bioavailable” may not be necessary here as a descriptor before introducing the concept in next paragraph

 Zinc bioavailability

· L 92-93 – consider: “Nutrient bioavailability is defined as the proportion of a CONSUMED nutrient in diet that is ULTIMATELY AVAILABLE TO BE utilized BY CELLS in the body, AFTER OVERCOMING BARRIERS IN THE GUT LUMEN AND ENTEROCYTE.” 

· L 95 – consider whether the term “distribution” may be more descriptive than “retention” (consider again in L 102 and in L 404)

· L 109-112 – Excellent point to have reinforced; (please check source for spelling surname of J Sam Godber in ref #28)

 3.1 Zinc luminal bioavailability FOR absorption, … 3.1.1 – 3.1.5

· Consider a brief mention of other influencing factors (eg, pH and gut transit time)

· L 126-130 – Breaking this lengthy sentence into 2-3 sentences may improve clarity of the important steps for the reader

· L 132 – does MW play a role in ligand solubility?

· L 148 – does transport velocity increase?

· L 152-153 – consider: “…acrodermatitis enteropathica (AE), a rare CONDITION CHARACTERIZED BY severe zinc deficiency…”

· L 156 – consider: “…two transport proteins, EACH WITH 8 TRANSMEMBRANE DOMAINS, form a dimer…”

· L 180 – the reader may appreciate completing the story on how ZIP4 recovery occurs following poor intake (KLF4, transcription, post-transcription, etc)

· L 249 – “(Caco-2 cells) [68].”

· L 343 – if available, provide molar ratios (Fe:Zn) for the reader to appreciate

  3.2  Zinc utilization

· L 402 – would it be true to say: “… no body zinc stores OF FREE ZINC OUTSIDE OF STRUCTURAL AND CATALYTIC INCORPORTION, absorbed zinc…”

· L 411 – consider: “…significant importance to the DISTRIBUTION AND cellular utilization…”

· Could refer to Table 1 that lists both albumin and MT

· L 467 – “…between zinc and FFA.”

 Thank you for this submission.

Author Response

Reviewer 2: 
Thank you for your work in zinc nutrition.  Appreciate the thorough review of an important physiologic topic.  The systemic bioavailability of zinc remains an important consideration both in public health and clinical therapy.  I offer some minor comments/suggestions for your consideration that may benefit the reader. 

Thank you for your thorough review and comments. We have responded point-by-point below. 

General Comments
1. Consider whether a brief description of the complexity of compartmental analysis for overall zinc kinetics would provide the reader some additional context for appreciating your review of bioavailability.

Compartmental modeling methodology and its use as a tool for understanding zinc kinetics is beyond the scope of this manuscript. We are working on a second manuscript that discusses the implications of these concepts on a larger scale, and plan to include further explanation of compartmental modeling there. 

2. Consider a brief discussion about zinc supplements, of the varying salts, with different potential bioavailability, which readers may appreciate.  The physiologic/pharmacologic concept of bioavailability remains similar: taking into account factors at sequential steps from pre-absorption (eg solubility), the absorption process across the enterocyte’s apical membrane (eg, transporters), disposition within enterocyte (eg, storage, efflux), post-absorption from basolateral membrane (eg, transporters, carriers, and distribution to sites of function or excretion).  May require reference to paracellular absorption at pharmacologic dosing (commonly 50 mg elemental zinc up to 2-3 times daily).

Consideration for the interest in supplementation is well received. A related term, drug bioavailability, is defined as the proportion of dose that reaches the site of action in its active form. A major distinction between nutrient and drug bioavailability is the presence of food. Zinc is naturally a component of food, although dietary zinc may be provided artificially as a fortificant, or as orally administered supplement pills. The latter may be taken fasted or with food. Other therapeutic forms, such as for topical administration or injection, do not interact directly with the gastrointestinal tract or digestive processes that determine nutrient bioavailability. Thus, either or both definitions could apply to zinc, primarily depending on whether the purpose is nutritional or therapeutic. Although we emphasize nutrient bioavailability and discuss roles of food components and digestive processes in enhancing or inhibiting zinc absorption, distribution and utilization; data relevant to the therapeutic administration of zinc are also discussed. 

3. Address limited roles of DMT1, ZIP10, ZIP11??

We discussed zinc transporters that are known or possible key regulators of zinc absorption. Other transporters are active in the transfer and tissue distribution of zinc after absorption by the intestinal mucosa. However, they represent a complex topic that is beyond the focus of this paper, and they are reviewed extensively by others (cited in 3.2).

 Introduction
· Line 24 – can remove “a” (…zinc deficiency affects numerous critical functions…)
Removed “a”

· L 27-28 – readers unfamiliar with nutrition status surveys may find confusing given the subsequent statement in Line 36-37 re: Zn deficiency
Reworded to distinguish estimates from true prevalence

· L 40-41 – good point about the focus to increase Zn [bio]availability; consider mentioning both diet and supplementation
Added “from diet or supplements”

Unique chemistry of zinc
· L 69 – can remove the first mention of “binding” (…metals with the lowest protein binding preference.”)
Removed first mention of “binding”

· L 72-73 – consider: “…movement of DIETARY OR SUPPLEMENTED zinc from its GUT absorption through to its delivery to SYSTEMIC cells…”
Reworded: “movement of zinc from its intestinal absorption, through to its delivery to systemic cells”

· L 79-80 – appreciated this illustrative example
Thank you.

· L 84 – using “bioavailable” may not be necessary here as a descriptor before introducing the concept in next paragraph
Removed “bioavailable”

Zinc bioavailability
· L 92-93 – consider: “Nutrient bioavailability is defined as the proportion of a CONSUMED nutrient in diet that is ULTIMATELY AVAILABLE TO BE utilized BY CELLS in the body, AFTER OVERCOMING BARRIERS IN THE GUT LUMEN AND ENTEROCYTE.” 
Refer to response to comment 2

· L 95 – consider whether the term “distribution” may be more descriptive than “retention” (consider again in L 102 and in L 404)
“Distribution” is added, however we also kept “retention” as it captures the implication that not all absorbed zinc stays in the body, and excretion back into the lumen is the primary point of regulation in humans. 

· L 109-112 – Excellent point to have reinforced; (please check source for spelling surname of J Sam Godber in ref #28)
Surname spelling corrected

3.1 Zinc luminal bioavailability FOR absorption, … 3.1.1 – 3.1.5
· Consider a brief mention of other influencing factors (eg, pH and gut transit time)
The role of gastric acidity in hydrolyzing zinc is mentioned in 3.1. Thus, differences in intragastric pH were proposed as a possible explanation for lower zinc absorption and greater variability from zinc oxide vs. zinc gluconate or citrate (Wegmuller 2014). Although peristaltic mobility and fluid transit time are known influences in intestinal absorption, especially with enteric recirculation [44], we are not aware that their contribution to zinc absorption or reabsorption has been measured. We noted this in the text. 

· L 126-130 – Breaking this lengthy sentence into 2-3 sentences may improve clarity of the important steps for the reader
Reworded

· L 132 – does MW play a role in ligand solubility?
Generally, larger molecules of the same type are less soluble. 

· L 148 – does transport velocity increase?
Transport velocity was not measured in the cited study. 

· L 152-153 – consider: “…acrodermatitis enteropathica (AE), a rare CONDITION CHARACTERIZED BY severe zinc deficiency…”
Added “condition characterized by”

· L 156 – consider: “…two transport proteins, EACH WITH 8 TRANSMEMBRANE DOMAINS, form a dimer…”
Added “each with 8 transmembrane domains”

· L 180 – the reader may appreciate completing the story on how ZIP4 recovery occurs following poor intake (KLF4, transcription, post-transcription, etc)
Zip4 is recovered following transcriptional regulation mediated by Krüppel-like factor 4. In murine models of zinc deficiency, increases in Zip4 transporter expression and localization to the plasma membrane, and zinc absorption, have been observed in response to low zinc intake. However, structural motifs responsible for zinc sensing differ between murine and human Zip4. Although currently available data do not rule out the upregulation of zinc absorption in humans, tracer data indicate that it is primarily in response to factors other than low zinc intake. Following a period of severe dietary zinc depletion in men, zinc balance was maintained via a decrease in endogenous zinc excretion, and a compensatory increase in zinc absorption was not observed following a previous period of low zinc intake. However, fractional zinc absorption increases in response to normal pregnancy.

· L 249 – “(Caco-2 cells) [68].”
Corrected

· L 343 – if available, provide molar ratios (Fe:Zn) for the reader to appreciate
Valberg, et al. 1984, reduced Zn absorption observed at Fe:Zn molar ratio of 10:1
Sandstrom, et al. 1985, reduced Zn absorption observed at Fe:Zn molar ratios of 1:1, 2.5:1, and 25:1

3.2  Zinc utilization
· L 402 – would it be true to say: “… no body zinc stores OF FREE ZINC OUTSIDE OF STRUCTURAL AND CATALYTIC INCORPORTION, absorbed zinc…”
There are no stores for labile zinc readily released in response to low intakes. 

· L 411 – consider: “…significant importance to the DISTRIBUTION AND cellular utilization…”
Added “distribution”

· Could refer to Table 1 that lists both albumin and MT
Added reference to Table 1

· L 467 – “…between zinc and FFA.”
Used “FFA”

Thank you for this submission.
Thank you for your comments!